# Comparative Analysis of the Nutritional and Sensory Profiles of Commercial Processed Meat Products Made from Beef and Plant-Based Protein

**DOI:** 10.3390/nu17111771

**Published:** 2025-05-23

**Authors:** Soomin Oh, Da Young Lee, Dongwook Kim, Yousung Jung, Sun Jin Hur, Aera Jang

**Affiliations:** 1Department of Applied Animal Science, Kangwon National University, Chuncheon 24341, Republic of Korea; osm808@naver.com (S.O.); donguk8282@naver.com (D.K.); dbtjd97@naver.com (Y.J.); 2Department of Animal Science and Technology, Chung-Ang University, Anseong 17546, Republic of Korea; laeah49@naver.com (D.Y.L.); hursj@cau.ac.kr (S.J.H.)

**Keywords:** beef, meat, processed meat, tteokgalbi, plant-based meat analogs, nutrition, nutritional labeling, sustainability, vegetarian, sensory characteristics

## Abstract

**Background:** Plant-based meat analogs (PBMAs) have attracted attention as alternatives to traditional meat. **Methods:** In this study, three beef products (BPs; BP-1, BP-2, BP-3) and three PBMA products (PPs; PP-1, PP-2, PP-3) purchased from a Korean market were evaluated for nutritional profile-based nutritional facts, such as proximate composition, total calories, and levels of mineral, cholesterol, sugar, fatty acids, and amino acids, and for sensory characteristics. **Results:** Cholesterol and vaccenic acid were detected only in the BP samples. The levels of crude lipids, zinc, palmitic acid, stearic acid, oleic acid, saturated fatty acids, monounsaturated fatty acids, threonine, and lysine, and the score for taste and flavor were higher for the BPs than for the PPs (*p* < 0.05). By contrast, the levels of carbohydrates, calcium, iron, magnesium, sodium, glucose, sucrose, total sugar, isoleucine, phenylalanine, and glutamic acid were higher for the PPs than for the BPs. The levels of protein, potassium, phosphorus, copper, aluminum, valine, leucine, histidine, and arginine did not differ significantly between the BPs and the PPs. Multivariate analysis indicated that palmitic acid influenced the differences in nutritional profiles between the BPs and the PPs. Additionally, discrepancies were observed between the measured and labeled values for total sugar in PP-1 and PP-3. **Conclusions:** These findings offer valuable insights for the development of processed meat products using beef and PBMAs and may help consumers make informed purchasing decisions through the provision of accurate and reliable nutritional information.

## 1. Introduction

Meat quality is determined by a variety of factors that affect customer preferences, including nutritional quality, safety, and sensory qualities [1]. The consumption of meat products has affected the evolution and development of humans as they are essential sources of protein and highly accessible minerals, such heme iron, zinc, and the B vitamins [2]. Red meat, such as beef, is generally recognized as a crucial source of essential nutrients for a well-balanced diet [3]. Food processing ensures the consistent availability of nutrition and flavor across seasons, and can improve product quality [4]. Tteokgalbi is a representative Korean processed meat product made by grinding beef cuts, shaping them into desired forms [5], and combining them with various vegetables, resembling a patty with seasoning. With increasing consumer demand for fresh convenience foods, processed meat products have become widely available, from supermarkets to convenience stores [6].

Global population growth and rising incomes are expected to drive a 70% increase in meat demand by 2050 [7]. In response, the food industry is moving toward the production of plant-based meat analogs (PBMAs) as substitutes for meat products, aiming to replicate the flavor, nutrition, and fibrous structures of traditional meat [8]. The PBMA market is projected to grow to USD 30.9 billion by 2026 and reach USD 85 billion by 2030 [9]. Common PBMAs include soy-based textured vegetable proteins (TVPs) [10]. However, plant-source proteins are often low in essential amino acids, such as lysine, methionine, and cysteine [11]. To address these nutritional deficiencies and to mimic the characteristics of meat products, various studies have focused on developing new PBMA sources through ingredient innovation [12]. For instance, to overcome the nutritional limitations of PBMAs, various pulse ingredients, including chickpea protein, faba bean protein, lentil protein, and mung bean protein, are being explored alongside conventional sources, such as pea protein, soy protein, and wheat protein [13].

According to the labeling standards of the Ministry of Food and Drug Safety (MFDS) [14] in Korea, nutritional labels must list calories, and the levels of sodium, carbohydrates, total sugars, lipids, trans and saturated fatty acids, cholesterol, and proteins, along with the percentage of daily nutritional values. The MFDS [14] also recommends that the actual measured values for calories, sodium, total sugars, trans and saturated fatty acids, lipids, and cholesterol should remain less than 120% of the labeled amount. Conversely, the values for carbohydrates, dietary fiber, protein, vitamins, and minerals should be at least 80% of the labeled amount. Although most PBMA products provide nutritional information, some processed beef products lack labeling, as they can be classified as processed meat products made or redistributed by the operators of livestock meat processing businesses [14]. These products are exempt from nutritional labeling under the rules for labeling and advertising of food and other products of the MFDS [15]. Accurate nutritional information should be provided to enable the consumers to make informed choices when purchasing meat products. Previous studies have investigated the nutritional composition of PBMAs and compared it with that of conventional fresh meat [8]. However, the nutritional composition of processed meat may differ from that of fresh meat due to the complex processing steps it undergoes [4].

Currently, comparative data on the nutritional labels and actual measured nutrient profiles of beef products (BPs) and PBMAs products (PPs), as well as the application of multivariate analysis to their nutritional composition, remain limited. Therefore, the aim of this study is to evaluate the nutritional characteristics of commercially available BPs and PPs by measuring their actual nutrient composition and comparing it with the labeled information, and to identify the similarities and differences between them through the application of multivariate analysis.

## 2. Materials and Methods

### 2.1. Sample Preparation

In this study, to select products that are easily accessible to consumers, three Hanwoo beef processed products (BP-1, BP-2, and BP-3) and three PBMA products (PP-1, PP-2, and PP-3), all in the form of tteokgalbi, were purchased from online shopping platforms in the Republic of Korea. The ingredient information for the BPs and the PPs are presented in Table 1. The products were stored in the laboratory at −18 °C until analysis. Before analyzing the nutritional profiles, samples of the products were collected and finely ground using a food mixer.

### 2.2. Proximate Composition and Total Calories

The proximate compositions of the samples were evaluated according to the methods of the Association of Official Agricultural Chemists (AOAC, 1997) [16]. The moisture content was determined by drying in the oven at 105 °C, while the crude protein content was assessed using the Kjeldahl method. The crude lipid content was measured via Soxhlet extraction using the ether (99.0% purity, Samchun Chemical Co., Ltd., Seoul, Republic of Korea), and crude ash content was determined by heating the sample in a furnace at 550 °C. The carbohydrate content was calculated by subtracting the total moisture, protein, lipids, and ash percentages from 100. The total calorie content of the samples was calculated by multiplying the protein and carbohydrate contents per gram of sample by 4 kcal/g and the lipid content by 9 kcal/g [1].

### 2.3. Mineral Contents Determination

To determine the mineral content of the samples, 2 g of each sample was heated in the furnace at temperatures ranging from 550 to 600 °C for 12 h, according to the method described by the Food Code of the MFDS (2024) [17]. After cooling to an ambient temperature, the samples were subjected to overnight digestion in 10 mL of HCl solution (35.0–37.0% purity, Samchun Chemical Co., Ltd., Seoul, Republic of Korea) and mixed with an equal volume of water (*v*/*v*). Subsequently, the digested samples were filtered through filter paper (Whatman No. 6, Whatman plc, UK). The mineral content in the filtrate was analyzed using an inductively coupled plasma emission spectrophotometer (OPTIMA 7300 DV, PerkinElmer, Shelton, CT, USA).

### 2.4. Cholesterol Content

The cholesterol content in the samples was analyzed using a slightly modified method based on the Food Code of MFDS (2024) [17]. An amount of 2 g of the sample were mixed with 10 mL of a saponification reagent, which consisted of potassium hydroxide (95.0% purity, Samchun Chemical Co., Ltd., Seoul, Republic of Korea) and ethanol (≥99.0% purity, Supelco™, Millipore Sigma Supelco, Bellefonte, PA, USA), and 1 mL of 5α-cholestane (97.0% purity, Sigma-Aldrich, St. Louis, MO, USA) solution, followed by homogenization for 15 s. The mixture was then incubated at 60 °C for 1 h. After the reaction was completed, the sample was cooled using cold water and subsequently mixed with 5 mL of distilled water and 10 mL of hexane (95.0% purity, Samchun Chemical Co., Ltd., Seoul, Republic of Korea). The mixture was centrifuged at 2000 rpm for 10 min to induce phase separation. An amount of 1 mL of the upper hexane layer was transferred to a 20 mL scintillation vial and evaporated to dryness under a fume hood. The dried residue was then mixed with 200 µL of pyridine (99.0% purity, Samchun Chemical Co., Ltd., Seoul, Republic of Korea) and 100 µL of Sylon BFT (99.0% purity, Supelco™, Millipore Sigma Supelco, Bellefonte, PA, USA), gently vortexed, and used for gas chromatography (GC) analysis. The derivatized samples were analyzed using GC (Agilent 8890 GC, Agilent Technologies, Inc., Santa Clara, CA, USA) equipped with an HP-5 column (30 m × 0.32 mm × 0.25 µm; Agilent Technologies, Inc.). The carrier gas was N_2_ with a flow rate of 1.5 mL/min and a split ratio of 5:1. The injector and flame ionization detector temperatures were set at 250 °C and 300 °C, respectively. The column temperature program was as follows: the initial temperature was held at 190 °C for 2 min, increased to 270 °C at a rate of 40 °C/min, and then maintained at 270 °C for 25 min. The cholesterol content was calculated as the ratio of the target peak area to the internal standard peak area and expressed as mg per 100 g of meat.

### 2.5. Sugar Content Determination

The sugar content of the samples was measured according to the method of the Food Code of the MFDS (2024) [17]. A total of 5 g of each sample were vortexed with 25 mL of ether (99.0% purity, Samchun Chemical Co., Ltd., Seoul, Republic of Korea). This process was repeated after centrifugation (783× *g*, 10 min, 4 °C) to separate the ethers. After completely evaporating the ether using N_2_ gas, distilled water (25 mL) was added and the weight was measured. To extract the sugar, the sample was boiled in a water bath for 25 min at 85 °C. After cooling the mixture to room temperature, the extraction solvent was added to restore the original weight. Subsequently, the mixture was filtered through a 0.45 μm nylon membrane filter and used as a test solution for high-performance liquid chromatography (Agilent Infinity 1260 series) with a refractive index detector. The operating conditions were as follows: mobile phase of acetonitrile (99.9% purity, Samchun Chemical Co., Ltd., Seoul, Republic of Korea) and water (70:30, *v*/*v*); flow rate of 1.4 mL/min; and injection volume of 15 µL. Fructose, glucose, sucrose, maltose, and lactose standards (purity ≥ 99.0% for fructose and maltose; ≥99.5% for glucose, sucrose, and lactose) were purchased from Sigma-Aldrich (St. Louis, MO, USA).

### 2.6. Fatty Acid Composition

The fatty acid composition of the samples was determined following the method of Lee et al. with slight modifications [18]. Briefly, 5 g of each sample was mixed with 50 µL of butylated hydroxytoluene (99.0% purity, Samchun Chemical Co., Ltd., Seoul, Republic of Korea). Subsequently, 30 mL of Folch solution, composed of chloroform (99.5% purity, Samchun Chemical Co., Ltd., Seoul, Republic of Korea) and methanol (99.9% purity, Samchun Chemical Co., Ltd., Seoul, Republic of Korea) in a 2:1 (*v*/*v*) ratio, was added, and the mixture was homogenized twice using a homogenizer. The sample was incubated at room temperature for 3 h with shaking every 30 min, followed by filtration. The filtrate was mixed with 9 mL of a 0.88% (*w*/*v*) sodium chloride solution (99.0% purity, Daejung Chemicals, Siheung, Republic of Korea) and shaken for 2 min. The sample was then left to stand for 24 h to induce the separation of the chloroform and the water layers. After separation, 10 mL of the lower layer was collected and dried to extract the lipids. Nitrogen was used during drying to prevent oxidation of the fatty acids. Chloroform (2 mL) was added to dissolve the extracted lipids, followed by the addition of 1 mL of a boron trifluoride–methanol solution (Sigma-Aldrich, St. Louis, MO, USA). Subsequently, the mixture was incubated at 60 °C for 40 min. After incubation, the mixture was cooled with cold water, and 3 mL of hexane (95.0% purity, Samchun Chemical Co., Ltd., Seoul, Republic of Korea) and 8 mL of distilled water were added. The mixture was centrifuged for 5 min at 2000 rpm and the upper layer was collected for fatty acid analysis using gas chromatography–mass spectrometry (GC-MS; Agilent Technologies, Inc., model 7890B) with a mass spectrometer (Agilent Technologies, Inc., model 5977B) and DB-WAX column (Agilent Technologies, Inc.). For fatty acid analysis, the GC-MS conditions were adapted from the method outlined by Lee et al. [18]. The fatty acid standard used for this analysis was a FAME mix (CRM18920; Supelco™, Millipore Sigma Supelco, Bellefonte, PA, USA).

### 2.7. Amino Acid Composition

The amino acid compositions of the samples were determined using the method described by Joo et al. [19]. Briefly, 1 g of each sample was homogenized for 15 s at 12,000 rpm using a homogenizer. Subsequently, 6 M HCl (35.0–37.0% purity, Samchun Chemical Co., Ltd., Seoul, Republic of Korea) was added to the homogenate in a glass bottle. Nitrogen was added to the mixture, which was then hydrolyzed for 24 h at 110 °C. Following hydrolysis, the mixture was transferred to a 50 mL volumetric flask and diluted with a sodium citrate buffer (pH 2.2) (Samchun Chemical Co., Ltd., Seoul, Republic of Korea) using a calibrated tube. The solution was filtered through a 0.45 µm membrane and placed in an auto-sampler vial. An amino acid analyzer (SYKAS433, Sykam GmbH, Eresing, Germany) was used to determine the amino acid composition.

### 2.8. Sensory Evaluation

A sensory evaluation of the BP and PP samples was performed by 125 panelists aged 20–50 years from the Kangwon National University and Chung-Ang University. The samples were cooked in a frying pan at a surface temperature of 180 °C until their internal temperature reached 72 ± 3 °C. They were then presented in portions measuring 2 × 2 × 1 cm. The panelists rinsed their mouths with water after each sample evaluation. The samples were evaluated for appearance, color, off-flavor, taste, flavor, juiciness, tenderness, and overall acceptability using a nine-point hedonic scale. The ratings for color, taste, flavor, and overall acceptability ranged from 1 (very bad) to 9 (very good). Off-flavor was rated from 1 (very weak) to 9 (very strong), while juiciness and tenderness were rated from 1 (very dry or very hard) to 9 (very juicy or very tender). This study was conducted in accordance with the Declaration of Helsinki, and the protocol was approved by the institutional review board of Kangwon National University (Approval No. 2023-11-004-002) on 10 January 2024. Verbal informed consent was obtained from the participants. Informed consent for participation was obtained from all subjects involved in this study.

### 2.9. Statistical Analysis

The BP-1, 2, 3 and PP-1, 2, 3 samples were analyzed in triplicate. A statistical analysis was conducted using the SAS software (version 9.4; SAS Institute, Cary, NC, USA) with one-way analysis of variance (ANOVA). Significant differences between the BPs and the PPs were determined using Tukey’s test. The data are presented as mean values with standard error of the mean (SEM). To compare the nutritional profiles of the BPs and the PPs, multivariate analysis was performed using partial least squares discriminant analysis (PLS-DA) and variable importance in projection scores (VIP scores) with log-transformed and auto-scaled data from MetaboAnalyst 6.0 (https://www.metaboanalyst.ca/). Statistical significance was set at *p* < 0.05.

## 3. Results and Discussion

### 3.1. Proximate Composition, Total Calories, and Cholesterol Content

The proximate composition and total calories of the BPs and the PPs are presented in Table 2. Proteins, which are crucial for brain growth and development, are mainly found in meat [9]. In this study, the crude protein of the PPs did not differ significantly from that of the BPs. This suggests that the protein content of the PPs can be similar to that of the BPs depending on its ingredient formulation. Hidayat et al. reported that, when beef was replaced with TVP in sausages, up to 40% substitution did not result in any significant change in the protein content [20]. The crude lipid content of the BPs was higher than that of the PPs, whereas the carbohydrate content of the BPs was lower than that of the PPs (*p* < 0.05). Bakhsh et al. reported that meat analog patties had lower fat content and higher crude fiber content than beef patties [9]. Fat plays a crucial role in the emulsification of meat products and contributes significantly to the improvement of texture and juiciness [21]. However, excess fat consumption is associated with adverse health effects [12]. Consequently, research on carbohydrate-based ingredients, including starch, carrageenates, and gellan gum, is being conducted to reduce the fat content of PBMAs while enhancing its juiciness, richness, and texture. In particular, starch is widely used in PBMAs because of its low cost, minimal impact on taste, and ability to improve texture [22]. However, an in vitro study showed that starch affects protein digestion, resulting in lower digestibility of PBMAs compared to that of real meat in the gastrointestinal phase [11]. Therefore, when manufacturing PBMAs, carbohydrate-based ingredients should be used while considering protein digestibility.

Excessive cholesterol is known to contribute to the development of cardiovascular diseases [23]. According to the food labeling standards of the MFDS (2023), the recommended daily intake of cholesterol is 300 mg [14]. In this study, cholesterol was not detected in the PPs, whereas the BPs contained 51.33 mg cholesterol/100 g product (Table 2). Similarly, Yang et al. reported that the cholesterol content in PBMA products was lower than 10 mg/100 g, whereas various beef cuts contained 48.40–63.47 mg/100 g cholesterol [8]. Moreover, Bohrer reviewed available studies on PBMAs and reported that the cholesterol content in PBMA products is typically lower than 10 mg/100 g [24]. These differences may be attributed to the higher cholesterol content in animal-based food than in the plant-based ingredients used in PBMAs [8]. In animals, cholesterol, the primary form of sterol, can be synthesized in all tissues, while its level is consistently regulated by the liver [23]. By contrast, plants contain phytosterols, which are sterols with bulky side chains containing methyl and/or ethyl groups [25]. However, cholesterol plays a crucial role in cellular function due to its unique structure, which consists of a proximal hydrophilic end, a distal hydrophobic end, and a central four-ring core [23]. Cholesterol is one of the main components of mammalian cells and helps them maintain biological function [25].

### 3.2. Mineral Contents

Minerals are essential dietary components required in small amounts to maintain good health throughout an individual’s life [26]. Beef is a particularly good source of zinc and iron for humans [27,28]. Zinc is a trace element that is involved in various biochemical functions, including cell growth, immune function, liver function, and antioxidant activity, and its deficiency is associated with various diseases [26]. In this study, the zinc content of the PPs was significantly lower than that of the BPs. However, the iron content of the PPs was significantly higher than that of the BPs (Table 3). Iron, mainly found in myoglobin and hemoglobin in meat, supports respiration and oxygen delivery to tissues [28]. Swing et al. reported that iron content in some commercial PBMA products is higher than that in ground beef [29]. Furthermore, this study showed that the calcium and magnesium contents in the PPs were significantly higher than those in the BPs, whereas the potassium, phosphorus, and copper levels in the PPs did not differ significantly from those in the BPs. These findings indicated that the manufacturers of PPs can incorporate nutritional fortifiers into the PP formula, thereby enabling the PPs to offer nutrients that closely resemble those found in the BPs [8]. Although PPs and BPs have comparable mineral content, the forms of certain minerals in meat differ from those in PBMAs. For example, there are two types of iron—non-heme iron from plant sources and heme iron from animal sources—with heme iron being absorbed more readily [30]. Hence, further research is required to examine the bioavailability of minerals in the BPs and the PPs relative to their mineral content.

Additionally, this study revealed that the PPs had a significantly higher sodium content than the BPs. Yang et al. reported that the sodium content of commercial PBMAs is higher than that of meat [8]. The flavor complexity of meat is attributed to its intricate compounds, whereas PBMAs often have a beany, bitter, and astringent taste. The ability to mimic the flavor of meat products is one of the challenges faced by the PBMA industry [7]. Although natural savory spices can mimic meaty aromas, salt remains a common ingredient to enhance taste [12]. However, consuming too much salt raises blood pressure and increases the risk of cardiovascular diseases [30]. Food labeling standards (MFDS, 2023) recommend a daily sodium intake of 2000 mg [14]. In this study, the sodium content in the PPs was 543.05 mg/100 g. Although these levels are below the daily limit, they may still benefit from reduction in line with current nutritional trends to maintain health [12].

### 3.3. Sugar Content

The sugar contents of the BPs and the PPs are presented in Table 4. PBMA ingredients often have undesirable flavors, such as off-flavor, astringency, and bitterness. These flavors stem from compounds such as glycosides, isoflavones, saponins, catechins, phenols, and phenolic acids [12]. To counteract these flavors, flavoring agents are frequently included in PBMA products [9]. Glucose is a Maillard reaction precursor that can produce meaty or roasted volatile compounds in PBMAs [7]. In this study, the glucose content in the PPs was significantly higher than that in the BPs (*p* < 0.05). Sucrose, a disaccharide commonly known as sugar, is the most widely utilized sweetening agent worldwide [31]. Here, the PPs had a significantly higher sucrose content than the BPs. Moreover, maltose was detected only in the BPs. A previous study has reported that maltose has cryoprotective effects and contributed to the stabilization of washed beef myofibrillar proteins during freezing [32].

The total sugar content of the PPs was higher than that of the BPs (*p* < 0.05). According to the food labeling standards of the MFDS (2023), the recommended daily intake of total sugar is 100 g [14]. In this study, the total sugar content of the PPs was 8.37 g/100 g. However, the margin of error between nutritional fact label values and actual measured values for total sugar content in PP-1 and PP-3 was 225% and 712%, respectively (Appendix A). According to the labeling standards for food (MFDS, 2023), the actual measured values of total sugar should be less than 120% of the labeled values [14]. The margin of error for total sugar content in PP-1 and PP-3 exceeded 120%, indicating a discord in the labeling standards for food.

### 3.4. Change of Fatty Acid Composition

The fatty acid composition of the BPs and the PPs are presented in Table 5. The primary fatty acids in the BPs were palmitic (C16:0), stearic (C18:0), and oleic acids (C18:1n9), whereas those in the PPs were lauric (C12:0), oleic (C18:1n9), and linoleic acids (C18:2n6). The difference in fatty acid composition between the BPs and the PPs may be attributed to the differences in the lipid source. Kim and Jang reported that palmitic, stearic, and oleic acids are the major fatty acids in Hanwoo beef, as oleic acid is abundant in the adipose tissues of cattle [27]. Yang et al. reported that fatty acids in coconut oil primarily include lauric, myristic, palmitic, and oleic acids, whereas sunflower and rapeseed oils primarily contain oleic and linolenic acids [8]. In this study, PP-1 included rapeseed, coconut, and sunflower oils, PP-2 included coconut oil, and PP-3 included sunflower oil.

In a previous study, palmitic and stearic acids in beef were found to correlate positively with sensory characteristics [33]. In addition, some reports have suggested that oleic acid increases the consumer preference for beef and that its consumption reduces the risk of metabolic diseases [34]. In this study, the palmitic, stearic, and oleic acid contents in all BPs were higher than those in all PPs (*p* < 0.05).

Meat is a major source of saturated fatty acids (SFAs), which can increase the risk of cardiovascular diseases and diabetes [35]. In this study, the SFA levels were significantly higher in the BPs than in the PPs. Nevertheless, the levels of monounsaturated fatty acids, which have been shown to lower the risk of cardiovascular disease, aid in weight management, and offer additional health advantages [36], were also significantly higher in the BPs than in the PPs. Trans fatty acids are known to cause adverse effects, such as cardiovascular disease and diabetes [35]. Vaccenic acid (C18:1n7) is the main trans fatty acid generated in the rumens of ruminant animals via the microbial hydrogenation of unsaturated fatty acids. Since 2003, the Food and Drug Administration has mandated the inclusion of trans fatty acids in nutritional labels. While conjugated linoleic acid is excluded from trans fat labels, vaccenic acid—unlike artificial trans fats—is not associated with negative cardiovascular implications. However, vaccenic acid has not been specifically differentiated from other trans fats. Therefore, some researchers have suggested separate quantification of vaccenic acid in animal products [37].

### 3.5. Change of Amino Acid Composition

Beef is an excellent protein source, and its amino acid composition not only enhances nutritional value but influences taste perception [19,27]. In this study, the levels of valine, leucine, histidine, and arginine levels did not differ significantly between the BPs and the PPs, while isoleucine and phenylalanine content in the PPs was significantly higher than that in the BPs (Table 6). These results may be related to the characteristics of the ingredients of PPs. The ingredients of PBMAs typically include soy, pea protein, and wheat gluten [12]. Although legumes and cereals contain limiting amino acids, an appropriate combination of plant-based proteins can improve the balance of amino acid composition and serve as a suitable alternative to animal-derived proteins [8]. However, we showed that the levels of threonine and lysine were significantly higher in the BPs than in the PPs (*p* < 0.05). These results are consistent with the findings of Yang et al., who reported that commercial PBMA products contain one or more limiting amino acids compared to real meat [8].

Amino acids can trigger intricate tastes. The umami taste of meat, which is attributed to the presence of glutamic and aspartic amino acids, plays a significant role in enhancing taste [19]. Therefore, it is crucial to mimic the presence of glutamic and aspartic amino acids in PBMAs. Samard and Ryu reported that the glutamic acid content in TVP is higher than that in beef [38]. In this study, we did not observe any significant difference in the aspartic acid levels of the BPs and the PPs, while the glutamic acid level was significantly higher in the PPs than in the BPs (Table 6). Furthermore, the levels of glycine and alanine, which contribute to the sweetness of meat [19], were significantly higher in the BPs than in the PPs.

### 3.6. Multivariate Analysis of Nutritional Profiles

Partial least squares discriminant analysis is a statistical method applied in food science, biostatistics, and medical research to generate reliable models even when there is high noise or multicollinearity among the independent variables owing to its ability to handle highly correlated predictors [39]. In this study, the sum of components 1 and 2 was 74.1%, which explained 74.1% of the total variance in product type in the nutritional profiles of the BPs and the PPs (Figure 1a). Component 1 exhibited a higher contribution rate than Component 2; based on Component 1, all BPs were distributed in the negative domain, whereas all PPs were distributed in the positive domain. These results revealed differences in the nutritional profiles of the BPs and the PPs. The effect of the product type was assessed using the VIP method, where elevated VIP scores identified the compounds that distinguished each treatment. When the scores exceed 1, they contribute to distinguishing clusters within treatment groups [39]. We observed that C16:0 (palmitic acid) contributed the most in distinguishing the clusters (Figure 1b).

### 3.7. Sensory Characteristics

The sensory characteristics of the BPs and the PPs are presented in Table 7. Appearance, color, and juiciness in the BP group were higher than those in the PP group (*p* < 0.05). However, the tenderness of the BPs and the PPs did not differ significantly. These results might be related to the characteristics of ingredients in PPs. Previous studies on the ingredients of PBMA designed to mimic real meat have suggested that PBMA patties formulated with commercial TVP and 3% methylcellulose represent a suitable combination for PBMA production when compared to beef patties [10]. However, in this study, the taste, flavor, and overall acceptability of the PPs were lower than those of the BPs, with a strong off-flavor observed in the PPs (*p* < 0.05). These results may be explained by differences in the nutritional composition. In meat products, fat enhances sensory acceptability [12]. In this study, crude fat was significantly higher in the BPs than in the PPs (Table 2). Furthermore, the fatty acid composition plays an important role in determining flavor profiles in food systems [39]. Cho et al. reported that in Hanwoo and Australian Angus beef, palmitic acid and stearic acid exhibited positive correlations with sensory attributes [33]. Also, oleic acid has been known to enhance flavor and increase the consumer preference for beef [34]. In this study, the levels of palmitic acid, stearic acid, and oleic acid were significantly higher in the BPs than in the PPs (Table 5). The amino acid composition of proteins also contributes to flavor characteristics [19]. Although no significant difference was observed in crude protein between the BPs and the PPs (Table 2), glutamic acid—known for its umami-enhancing properties—was significantly higher in the PPs, while aspartic acid levels did not differ significantly between the two groups (Table 6). Flavor-enhancing components were also analyzed in this study. Both sodium and total sugar contents were significantly higher in the PPs than in the BPs (Table 3 and Table 4). Despite the presence of these favorable flavor-enhancing nutrients in the PPs, sensory evaluation revealed that the BPs scored higher in flavor-related attributes. The results may be attributed to differences in raw material composition. Plant-based ingredients often contain undesirable compounds, such as glycosides, phenols, and phenolic compounds, which contribute to inherent off-flavors characterized by bitterness and astringency [12]. Hernadez et al. reported that patties made from PBMAs—such as Beyond Burger and Impossible Burger—exhibited weaker beef flavor intensity compared to conventional ground beef, but displayed stronger umami, nutty, smoky-charcoal, and musty/earthy notes. These sensory differences were associated with variations in the concentrations of flavor-active compounds, including pyrazines, furans, ketones, alcohols, and aldehydes [40]. These results highlight the challenges in mimicking the flavor of the BPs, and the need for flavor development in the PPs. To enhance the flavor of PPs, the activity of lipoxygenase, which is responsible for the formation of off-flavors during the heating of soybean protein, should be inhibited, or additives, such as β-cyclodextrin, phospholipase A2, and monosodium glutamate monohydrate, should be incorporated to mitigate these off-flavors [41,42].

## 4. Conclusions

In this study, we assessed the nutritional profiles and sensory characteristics of BPs and PPs. Cholesterol and vaccenic acid were detected only in the BP group. The levels of lipid, zinc, palmitic acid, stearic acid, oleic acid, MUFAs, threonine, and lysine, and the preference for taste and flavor were significantly higher in the BP group than in the PP group. However, the characteristics of some of the nutrients in the PPs were similar to those in the BPs. This implied that the PPs may have nutritional characteristics similar to those of the BPs. In addition, the total sugar content of PP-1 and PP-3 differed between the actual measured values and the nutritional facts. Currently, the nutritional characteristics of commercially available PPs do not completely match those of BPs. This indicates a need for further improvement to better mimic the qualities of BPs. These findings offer valuable insights for developing the meat product industry and promoting the production of sustainable food. However, this study has several limitations. First, the number of commercially available BPs and PPs evaluated was limited, and further analyses of a wider range of market products are necessary. Second, although the BPs showed superior sensory attributes related to flavor compared to the PP group, its higher content of fat, SFA, and cholesterol should be considered, and additional studies are needed to assess the potential health implications of BP and PP consumption. Third, the sensory evaluation was conducted by a specific group of panelists; therefore, to generalize the sensory characteristics of BPs and PPs, further studies incorporating a broader range of products, cooking methods, and diverse panelist groups are required.

## Figures and Tables

**Figure 1 nutrients-17-01771-f001:**
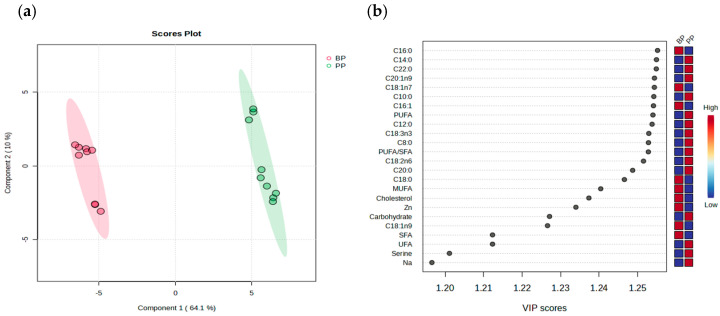
Multivariate analysis of nutritional profiles of the commercial processed meat products made from beef and plant-based meat analogs. Partial least squares discriminant analysis (PLS-DA) (**a**), and its variable importance in projection scores (VIP scores) (**b**) from beef products and plant-based meat analogs products; BP, beef product; PP, plant-based meat analogs product.

**Table 1 nutrients-17-01771-t001:** The listed ingredients and nutritional facts of the commercial processed meat products from beef and plant-based meat analogs.

Samples	The Listed Ingredients of the Products	Nutritional Facts (per 100 g)
BP-1	Hanwoo beef, dry bread crumbs, onions, sugar, pear juice, carrots, minced garlic, large green onions, potato starch, sesame oil, water, soju, salt, L-glutamate, black pepper, sesame seeds	Not shown
BP-2	Hanwoo beef, Hanwoo fat, rice cakes, moist bread crumbs, bulgogi sauce [soy sauce {amino acid liquid, brewed soy sauce extract, salt, garlic, green onions, fructose, pear puree (pear, vitamin C), onions, apple puree, seasoning sauce extract, black, pepper, citric acid, xanthan gum, caramel colorant}]	Not shown
BP-3	Hanwoo beef, sesame oil, corn syrup, brown sugar, garlic, onions, salt, soy sauce, potato starch, L-glutamate, caramel colorant, ginger, black pepper	Not shown
PP-1	Water, soy protein concentrates, rapeseed oil, sugar, coconut oil, onion, garlic extract, large green onions, methylcellulose, grill sauce, Cookmate NF, sesame oil, isolated soy protein, soy sauce powder, salt, gellan gum, isolated soy protein, yeast extract, rice flour, black pepper, food additive 1 [brown rice oil, refined processed oil, sunflower oil, flavoring 1, flavoring 2, d-alpha-tocopherol, propylene glycol], cocoa colorant, beet red, oat dietary fiber, cysteine hydrochloride, citric acid, food additive 2 [smoke oil, soybean oil], food additive 3 [tomato colorant, glycerin, caramel colorant]	Calories (255 kcal), sodium (530 mg), total carbohydrate (16 g), total sugar (5 g), lipid (15 g), trans fatty acids (0 g), saturated fatty acids (5 g), cholesterol (0 mg), protein (14 g)
PP-2	Processed bean products [vital wheat gluten, isolated soy protein, defatted soybean powder, starch, other processed ingredients], coconut oil, sauce [sugar, onion, hydrolyzed soy sauce, apple puree, brewed soy sauce], onion, large green onions, sugar, garlic, grain products, methylcellulose, seasoning, salt, sesame oil, yeast extract, food additive (dextrin, vegetable oil, modified starch, synthetic flavoring), processed sugar product, black pepper	Calories (255 kcal), sodium (510 mg), total carbohydrate (21 g), total sugar (12 g), lipid (15 g), trans fatty acids (0 g), saturated fatty acids (12 g), cholesterol (0 mg), protein (9 g)
PP-3	Sunflower oil, processed bean products, wheat gluten, seasoning, isolated soy protein, sauce (paste for tteokgalbi), soy sauce, onion, rice cakes, garlic, sugar, alcoholic beverages, cocoa colorant, pepper	Calories (205 kcal), sodium (680 mg), total carbohydrate (13 g), total sugar (1 g), lipid (10 g), trans fatty acids (0 g), saturated fatty acids (1.1 g), cholesterol (0 mg), protein (15 g)

BP-1, beef product-1; BP-2, beef product-2; BP-3, beef product-3; PP-1, plant-based meat analog product-1; PP-2, plant-based meat analog product-2; PP-3, plant-based meat analog product-3.

**Table 2 nutrients-17-01771-t002:** Proximate composition, total calories, and cholesterol content of the commercial processed meat products from beef and plant-based meat analogs.

Traits	BP	PP	SEM
Moisture (%)	54.22	54.29	0.758
Crude protein (%)	13.04	13.65	0.885
Crude lipids (%)	22.26 ^a^	11.14 ^b^	1.337
Carbohydrates (%)	9.13 ^b^	18.74 ^a^	0.371
Crude ash (%)	1.34 ^b^	2.18 ^a^	0.149
Total calories (kcal/100 g)	289.05 ^a^	229.85 ^b^	9.056
Cholesterol (mg/100 g)	51.33	N.D.	-

^a,b^ Means within a row with different superscripts differ significantly at *p* < 0.05. BP, beef product; PP, plant-based meat analog product; SEM, standard error of the mean. The content of carbohydrates was calculated as follows: 100 − (Moisture + Protein + Ash + Lipids).

**Table 3 nutrients-17-01771-t003:** Mineral contents of the commercial processed meat products from beef and plant-based meat analogs.

Minerals (mg/100 g)	BP	PP	SEM
Ca	25.54 ^b^	75.63 ^a^	8.338
Fe	1.20 ^b^	1.56 ^a^	0.117
K	199.47	210.67	24.047
Mg	19.31 ^b^	39.12 ^a^	3.646
Na	271.63 ^b^	543.05 ^a^	15.322
Zn	3.08 ^a^	1.11 ^b^	0.067
P	184.62	213.69	18.911
Cu	0.32	0.10	0.077
Al	0.17	1.16	0.338

^a,b^ Means within a row with different superscripts differ significantly at *p* < 0.05. BP, beef product; PP, plant-based meat analog product; SEM, standard error of the mean.

**Table 4 nutrients-17-01771-t004:** Sugar content of the commercial processed meat products from beef and plant-based meat analogs.

Sugar (g/100 g)	BP	PP	SEM
Fructose	0.10	0.23	0.053
Glucose	0.23 ^b^	1.30 ^a^	0.280
Sucrose	2.15 ^b^	6.83 ^a^	0.807
Maltose	0.52	N.D.	-
Lactose	N.D.	N.D.	-
Total	3.01 ^b^	8.37 ^a^	0.594

^a,b^ Means within a row with different superscripts differ significantly at *p* < 0.05. BP, beef product; PP, plant-based meat analog product; N.D., not detected; SEM, standard error of the mean.

**Table 5 nutrients-17-01771-t005:** Fatty acid composition of the commercial processed meat products from beef and plant-based meat analogs.

Fatty Acid Composition (%)	BP	PP	SEM
C8:0 (Caprylic acid)	0.20 ^b^	5.18 ^a^	0.064
C10:0 (Capric acid)	0.12 ^b^	3.35 ^a^	0.031
C12:0 (Lauric acid)	1.74 ^b^	10.22 ^a^	0.092
C14:0 (Myristic acid)	2.22 ^b^	6.94 ^a^	0.037
C16:0 (Palmitic acid)	25.54 ^a^	8.19 ^b^	0.119
C16:1 (Palmitoleic acid)	2.54 ^a^	0.32 ^b^	0.022
C18:0 (Stearic acid)	17.00 ^a^	6.26 ^b^	0.235
C18:1n9 (Oleic acid)	39.00 ^a^	30.84 ^b^	0.318
C18:1n7 (Vaccenic Acid)	1.37	N.D.	-
C18:2n6 (Linoleic acid)	6.15 ^b^	17.11 ^a^	0.166
C18:3n3 (α-Linolenic acid)	3.00 ^b^	8.19 ^a^	0.066
C20:0 (Arachidic acid)	0.34 ^b^	1.00 ^a^	0.013
C20:1n9 (Eicosenoic acid)	0.66 ^b^	1.77 ^a^	0.010
C20:3n3 (Eicosatrienoic acid)	0.06	N.D.	-
C20:4n6 (Arachidonic acid)	0.04	N.D.	-
C22:0 (Behenic acid)	N.D.	0.60	-
SFA	47.17 ^a^	41.76 ^b^	0.260
UFA	52.83 ^b^	58.24 ^a^	0.260
MUFA	43.58 ^a^	32.93 ^b^	0.300
PUFA	9.25 ^b^	25.31 ^a^	0.166
MUFA/SFA	0.92 ^a^	0.79 ^b^	0.011
PUFA/SFA	0.20 ^b^	0.61 ^a^	0.005

^a,b^ Means within a row with different superscripts differ significantly at *p* < 0.05. BP, beef product; PP, plant-based meat analog product; SFA, saturated fatty acids; UFA, unsaturated fatty acids; MUFA, monounsaturated fatty acids; PUFA, polyunsaturated fatty acids; N.D., not detected; SEM, standard error of the mean.

**Table 6 nutrients-17-01771-t006:** Amino acid composition of the commercial processed meat products from beef and plant-based meat analogs.

Amino Acid (% in Protein)	BP	PP	SEM
Aspartic acid	9.16	10.18	0.067
Threonine	3.22 ^a^	2.73 ^b^	0.023
Serine	2.52 ^b^	3.58 ^a^	0.024
Glutamic acid	17.93 ^b^	23.96 ^a^	0.210
Proline	6.28	6.83	0.098
Glycine	8.84 ^a^	4.37 ^b^	0.122
Alanine	7.33 ^a^	4.69 ^b^	0.050
Valine	5.53	5.64	0.035
Isoleucine	4.67 ^b^	5.16 ^a^	0.033
Leucine	7.89	8.35	0.052
Tyrosine	1.51	1.42	0.013
Phenylalanine	4.80 ^b^	5.78 ^a^	0.036
Histidine	4.87	4.79	0.034
Lysine	8.57 ^a^	5.69 ^b^	0.048
Arginine	6.80	6.83	0.045

^a,b^ Means within a row with different superscripts differ significantly at *p* < 0.05. BP, beef product; PP, plant-based meat analog product; SEM, standard error of the mean.

**Table 7 nutrients-17-01771-t007:** Sensory evaluation of the commercial processed meat products from beef and plant-based meat analogs.

Traits	BP	PP	SEM
Appearance	7.42 ^a^	6.59 ^b^	0.073
Color	7.37 ^a^	6.62 ^b^	0.076
Off-flavor	2.80 ^b^	4.08 ^a^	0.105
Taste	7.14 ^a^	4.81 ^b^	0.094
Flavor	6.90 ^a^	4.72 ^b^	0.096
Juiciness	5.82 ^a^	4.89 ^b^	0.091
Tenderness	6.09	6.17	0.083
Overall acceptability	7.13 ^a^	4.93 ^b^	0.087

^a,b^ Means within a row with different superscripts differ significantly at *p* < 0.05. BP, beef product; PP, plant-based meat analog product; SEM, standard error of the mean. Appearance, color, taste, flavor, overall acceptability (1 = very bad, 9 = very good), off-flavor (1 = very weak, 9 = very strong), juiciness (1 = very dry, 9 = very juicy), and tenderness (1 = very hard, 9 = very tender).

## Data Availability

The data presented in this study are available upon request from the corresponding author due to ethical reasons.

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
