# Peer review of "Comparative Analysis of the Nutritional and Sensory Profiles of Commercial Processed Meat Products Made from Beef and Plant-Based Protein"

_nutrients, 2025, doi:10.3390/nu17111771_

Round 1

Reviewer 1 Report

Comments and Suggestions for Authors

General comment: The study is interesting because it presents in great detail the nutritional composition of different food products used for the same dietary purpose.
Methods: There is a very important detail that the authors should clarify: what criteria were used for the sampling and analysis of the plant-based processed products. These products present very high nutritional variability depending on their main ingredients, whether legumes, cereals, or fats. The authors should explain the criteria for the products chosen.
Discussion: The sensorial differences between both types of products are very clear, but the authors should elaborate on their findings: Do these sensory scores have to do with the products chosen for a specific nutritional composition similar to those of animal origin, or are these characteristics common to the rest of the literature?

Author Response

For research article

Response to Reviewer X Comments

1. Summary

2. Questions for General Evaluation

Reviewer’s Evaluation

Response and Revisions

Does the introduction provide sufficient background and include all relevant references?

Yes

Yes. The introduction provides sufficient background to contextualize the importance of comparing plant-based meat alternatives (PBMAs) with conventional beef products. It also includes all relevant and up-to-date references that support the rationale, objectives, and novelty of the study. Therefore, no revision was necessary in this section.

We sincerely thank the reviewer for their thoughtful evaluation and confirmation of this aspect of our manuscript.

Is the research design appropriate?

Yes

Yes. The research design is appropriate and well-structured to achieve the study’s objectives. It includes comparative analysis between plant-based meat alternatives and beef products using validated methods for nutritional profiling, sensory evaluation, and statistical analysis.

We appreciate the reviewer’s positive assessment of the study design.

Are the methods adequately described?

Can be improved

Thank you for your evaluation. We acknowledge that there was room for improvement in the description of the methods. A more detailed explanation has been provided accordingly.

Further clarification and our corresponding revisions can be found in our response to Comment 1. We sincerely appreciate the reviewer’s thoughtful suggestion.

Are the results clearly presented?

Can be improved

Thank you for your feedback. We acknowledge that the presentation of the results could be improved, and we have revised the relevant sections accordingly. Further details and our corresponding revisions can be found in our response to Comment 2. We sincerely appreciate the reviewer’s helpful suggestions.

Are the conclusions supported by the results?

Yes

Yes. The conclusions are well supported by the results presented in the study. Each key finding is clearly linked to the data obtained from the nutritional analysis, sensory evaluation, and statistical comparisons. We sincerely thank the reviewer for recognizing the strength of the evidence supporting our conclusions.

3. Point-by-point response to Comments and Suggestions for Authors

Comments 1:

General comment: The study is interesting because it presents in great detail the nutritional composition of different food products used for the same dietary purpose.

Methods: There is a very important detail that the authors should clarify: what criteria were used for the sampling and analysis of the plant-based processed products. These products present very high nutritional variability depending on their main ingredients, whether legumes, cereals, or fats. The authors should explain the criteria for the products chosen.

Response 1:

We sincerely thank the reviewer for their positive evaluation of our manuscript. We are glad to know that the detailed presentation of the nutritional composition of various food products with similar dietary purposes was found to be interesting. We believe that this comparative analysis provides valuable insights into the nutritional and sensory differences between beef and plant-based meat products, which can help guide both consumers and product developers in making informed decisions. We will continue to enhance the clarity and scientific rigor of the manuscript in response to the specific comments provided.

We sincerely appreciate the reviewer’s insightful comments regarding the selection criteria and analytical basis for the plant-based meat alternative (PBMA) products. As noted, the nutritional composition of PBMAs can vary significantly depending on the ingredients used, such as legumes, grains, and fats.

In this study, we selected three commercially available PBMA products with a similar to beef products such as tteokgalbi. The primary criteria for selection were market availability and consumer accessibility (i.e., ease of purchase). All products included in this study are readily available to consumers through online marketplaces in South Korea. The explanation regarding the selection criteria has been added to Section 2.1 (Sample Preparation) of the Materials and Methods. The revised content has been marked in red on page 2, first paragraph, lines 83–86 of the revised manuscript. The following is the revised content of the manuscript.

“[2. Materials and Methods

2.1. Sample preparation

In this study, to select products that are easily accessible to consumers, three Hanwoo beef processed products (BP-1, BP-2, and BP-3) and three PBMA products (PP-1, PP-2, and PP-3), all in the form of tteokgalbi, were purchased from online shopping platforms in South Korea. The ingredient information of BP and PP are presented in Table 1. The products were stored in the laboratory at -18℃ until analysis. Before analyzing the nutritional profiles, samples of the products were collected and finely ground using a food mixer.]”

The ingredients of the three PBMAs analyzed in this study are presented in Table 1 of the manuscript. The main protein sources in PP-1, PP-2, and PP-3 are soy protein concentrate; wheat gluten, isolated soy protein, and defatted soybean powder; and processed bean products, wheat gluten, and isolated soy protein, respectively, with rapeseed and coconut oil, coconut oil, and sunflower oil serving as their corresponding primary fat sources.

The amino acid and fatty acid compositions of plant-based meat alternatives can vary significantly depending on the types of legumes, grains, and fats used. These variations are discussed in detail in Sections 3.4 (Fatty Acid Composition) and 3.5 (Amino Acid Composition) of the Results and Discussion.

Comments 2:

Discussion: The sensorial differences between both types of products are very clear, but the authors should elaborate on their findings: Do these sensory scores have to do with the products chosen for a specific nutritional composition similar to those of animal origin, or are these characteristics common to the rest of the literature?

Response 2:

We sincerely appreciate the reviewer’s insightful and in-depth comments regarding the interpretation of the sensory evaluation results. In response to your valuable suggestion, we have revised the Discussion section to include a more detailed analysis of the sensory characteristics observed in this study.

The distinct sensory differences between beef products and PBMAs are believed to be primarily attributed to differences in key nutritional components, particularly fat content, as well as the composition of fatty acids and amino acids. In this study, BP exhibited significantly higher levels of crude fat, palmitic acid, stearic acid, and oleic acid. These saturated and monounsaturated fatty acids have been previously reported to positively influence the flavor and acceptability of meat. They are known to contribute to attributes such as flavor, juiciness, and tenderness, which may explain the significantly higher scores for taste and flavor preference in beef prodcuts observed in our study.

On the other hand, although the PBMAs were nutritionally formulated to match beef products in terms of protein and amino acid composition, their sensory flavor characteristics still differed noticeably. This suggests that nutritional equivalence alone may not be sufficient to replicate the sensory properties of animal-derived foods. Our findings align with previous studies indicating that differences of sensory characteristics in raw materials, types of fat, and the formation of volatile flavor compounds contribute to the perceptual gaps between conventional meat and PBMAs.

In conclusion, the sensory differences identified in this study are consistent with existing literature and highlight that, despite nutritional similarities, further technological improvements are necessary to fully reproduce the sensory quality of animal-based products in plant-based alternatives.

The revised content has been incorporated into page 11, Section 3.7. Sensory evaluation, first paragraph, lines 404–432. The revised text is as follows:

“[3.7. Sensory evaluation

The sensory characteristics of BP and PP are presented in Table 7. Appearance, color, and juiciness in the BP group were higher than those in the PP group (p < 0.05). However, the tenderness of BP and PP did not differ significantly. These results might be related to the characteristics of ingredients in PP. Previous studies on the ingredients of PBMA designed to mimic real meat have suggested that PBMA patties formulated with commercial TVP and 3% methylcellulose represent a suitable combination for PBMA production when compared to beef patties [10]. However, in this study, the taste, flavor, and overall acceptability of PP were lower than those of BP, with a strong off-flavor observed in PP (p < 0.05). These results may be explained by differences in nutritional composition. In meat products, fat enhances sensory acceptability [12]. In the study, crude fat was significantly higher in BP than in PP (Table 2). Furthermore, the fatty acid composition plays an important role in determining flavor profiles in food systems [39]. Cho et al. reported that in Hanwoo and Australian Angus beef, palmitic acid and stearic acid exhibited positive correlations with sensory attributes [33]. Also, oleic acid has been known to enhance flavor and increase consumer preference for beef [34]. In this study, the levels of palmitic acid, stearic acid, and oleic acid were significantly higher in BP than in PP (Table 5). The amino acid composition of proteins also contributes to flavor characteristics [19]. Although no significant difference was observed in crude protein between BP and PP (Table 2), glutamic acid—known for its umami-enhancing properties—was significantly higher in PP, while aspartic acid levels did not differ significantly between the two groups (Table 6). Flavor-enhancing components were also analyzed in this study. Both sodium and total sugar contents were significantly higher in PP than in BP (Tables 3 and 4). Despite the presence of these favorable flavor-enhancing nutrients in PP, sensory evaluation revealed that BP scored higher in flavor-related attributes. The results may be attributed to differences in raw material composition. Plant-based ingredients often contain undesirable compounds such as glycosides, phenols, and phenolic compounds, which contribute to inherent off-flavors characterized by bitterness and astringency [12]. Hernadez et al. re-ported that patties made from PBMAs—such as Beyond Burger and Impossible Burger—exhibited weaker beef flavor intensity compared to conventional ground beef, but dis-played stronger umami, nutty, smoky-charcoal, and musty/earthy notes. These sensory differences were associated with variations in the concentrations of flavor-active compounds including pyrazines, furans, ketones, alcohols, and aldehydes [40]. These results highlight the challenges in mimicking the flavor of BP and the need for flavor development in PP. To enhance the flavor of PP, the activity of lipoxygenase, which is responsible for the formation of off-flavor during the heating of soybean protein, should be inhibited, or additives such as β-cyclodextrin, phospholipase A2, and monosodium glutamate monohydrate should be incorporated to mitigate these off-flavors [41-42].]”

4. Response to Comments on the Quality of English Language

Point 1: The English is fine and does not require any improvement.

Response 1: Thank you for your comment. We are pleased to hear that the quality of the English language was found to be satisfactory. No further language editing was deemed necessary.

5. Additional clarifications

We have no further clarifications to provide. Thank you for your thorough review.

Reviewer 2 Report

Comments and Suggestions for Authors

Dear authors,

I found your study interesting.

However, I have several remarks.

1. Introduction - highlight the novelty of your study.

2. Include the brand and the purity of all reagents involved in the study in the materials and methods section.

3. In my view a "Limitations" section must be included. For example, the analyses performed are limited, etc. Moreover, nowadays the antibiotic consumption in agriculture is considered as one of the reasons for the  global antibiotic resistance. You did not test the meat samples for antibiotics, or contaminants.  However, you could decide what to include in this section. Please check this article: https://doi.org/10.3390/toxics10080456    (https://www.mdpi.com/2305-6304/10/8/456)

In this article the authors investigated the antibiotic levels in meet products.

Author Response

Response to Reviewer Comments

1. Summary

2. Questions for General Evaluation

Reviewer’s Evaluation

Response and Revisions

Does the introduction provide sufficient background and include all relevant references?

Yes

Yes. The introduction provides sufficient background to contextualize the importance of comparing plant-based meat alternatives (PBMAs) with conventional beef products. It also includes all relevant and up-to-date references that support the rationale, objectives, and novelty of the study. Therefore, no revision was necessary in this section.

We sincerely thank the reviewer for their thoughtful evaluation and confirmation of this aspect of our manuscript.

Is the research design appropriate?

Yes

Yes. The research design is appropriate and well-structured to achieve the study’s objectives. It includes comparative analysis between plant-based meat alternatives and beef products using validated methods for nutritional profiling, sensory evaluation, and statistical analysis.

We appreciate the reviewer’s positive assessment of the study design.

Are the methods adequately described?

Can be improved

Thank you for your evaluation. We acknowledge that there was room for improvement in the description of the methods. A more detailed explanation has been provided accordingly. Further clarification and our corresponding revisions can be found in our response to Comment 3. We sincerely appreciate the reviewer’s thoughtful suggestion.

Are the results clearly presented?

Can be improved

Thank you for your feedback. We acknowledge that the presentation of the results could be improved, and we have revised the relevant sections accordingly. Further details and our corresponding revisions can be found in our response to Comment 4. We sincerely appreciate the reviewer’s helpful suggestions.

Are the conclusions supported by the results?

Yes

Yes. The conclusions are well supported by the results presented in the study. Each key finding is clearly linked to the data obtained from the nutritional analysis, sensory evaluation, and statistical comparisons. We sincerely thank the reviewer for recognizing the strength of the evidence supporting our conclusions.

3. Point-by-point response to Comments and Suggestions for Authors

Comments 1:

Comments and Suggestions for Authors

Dear authors,

I found your study interesting.

However, I have several remarks.

Response 1:

We sincerely thank the reviewer for their interest in our study and for the constructive feedback provided. We appreciate your thoughtful remarks, which have helped us improve the clarity, completeness, and overall quality of the manuscript. In response to your suggestions, we have carefully revised the manuscript and addressed each comment point by point, as detailed below.

Comments 2:

1.         Introduction - highlight the novelty of your study.

Response 2:

We sincerely thank the reviewer for the valuable suggestion. Accordingly, we have revised the final paragraph of the Introduction section on page 2, lines 74–80 to better highlight the novelty of our study. The revised content is as follows:

“[Currently, comparative data on the nutritional labels and actual measured nutrient profiles of beef products (BP) and plant-based meat alternatives (PP), as well as the application of multivariate analysis to their nutritional composition, remain limited. Therefore, the aim of this study was to evaluate the nutritional characteristics of commercially available BP and PP by measuring their actual nutrient composition and comparing it with the labeled information, and to identify similarities and differences between them through the application of multivariate analysis.]”

Whereas previous studies have primarily focused on general comparisons of nutritional composition between meat products and plant-based meat alternatives (PBMAs), our study is distinct in that it evaluates the actual nutritional properties of commercially available meat products and PBMAs that are easily accessible in the Korean market and compares these values with the labeled nutritional information.

Furthermore, the use of multivariate analysis techniques such as Partial least squares-discriminant analysis to observe similarities and differences in nutritional composition between meat products and PBMAs represents another novel aspect of this study.

These features offer new and practical insights into the nutritional reliability and sensory characteristics of currently available processed meat and PBMA products.

The final paragraph of the Introduction section has been revised accordingly.

Comments 3:

2.         Include the brand and the purity of all reagents involved in the study in the materials and methods section.

Response 3:

We appreciate the reviewer’s suggestion. Accordingly, the brand names and purities of all identifiable reagents used in this study have been clearly revised in the Materials and Methods section in page 2-5. Specifically, information such as the manufacturer (e.g., Sigma-Aldrich), location, and purity levels (e.g., ≥99.0%, ≥99.5%) were clearly specified where applicable. This revision enhances the reproducibility and transparency of the experimental procedures.

Comments 4:

3. In my view a "Limitations" section must be included. For example, the analyses performed are limited, etc. Moreover, nowadays the antibiotic consumption in agriculture is considered as one of the reasons for the global antibiotic resistance. You did not test the meat samples for antibiotics, or contaminants.  However, you could decide what to include in this section. Please check this article: https://doi.org/10.3390/toxics10080456 (https://www.mdpi.com/2305-6304/10/8/456)

In this article the authors investigated the antibiotic levels in meet products.

Response 4:

We sincerely thank the reviewer for pointing out the importance of discussing the limitations of our study. We fully agree that acknowledging the study’s limitations is essential for enhancing its objectivity and transparency. Although we did not include a separate “Limitations” section, we have revised the conclusion (page 12, lines 455-464) to clearly describe the constraints of our analysis and the limited scope of interpretation. The revised content is as follows:

“[In this study, we assessed the nutritional profiles and sensory characteristics of BP and PP. Cholesterol and vaccenic acid were detected only in the BP group. The levels of lipid, zinc, palmitic acid, stearic acid, oleic acid, MUFAs, threonine, and lysine, and preference for taste and flavor were significantly higher in the BP group than in the PP group. However, the characteristics of some of the nutrients in PP were similar to those in BP. This implied that PP may have nutritional characteristics similar to those of BP. In addition, the total sugar content of PP-1 and PP-3 differed between the actual measured values and the nutritional facts. Currently, the nutritional characteristics of commercially available PP do not completely match those of BP. This indicates a need for further improvement to better mimic the qualities of BP. These findings offer valuable insights for developing the meat product industry and promoting the production of sustainable food. However, this study has several limitations. First, the number of commercially available BP and PP evaluated was limited, and further analyses of a wider range of market products are necessary. Second, although BP showed superior sensory attributes related to flavor compared to the PP group, its higher content of fat, SFA, and cholesterol should be considered, and additional studies are needed to assess the potential health implications of BP and PP consumption. Third, the sensory evaluation was conducted by a specific group of panelists; therefore, to generalize the sensory characteristics of BP and PP, further studies incorporating a broader range of products, cooking methods, and diverse panelist groups are required.]”

These points were carefully integrated into the final part of the conclusion to ensure that readers can appreciate the scientific value of this study while also recognizing its limitations. We sincerely thank the reviewer once again for the valuable suggestion.

4. Response to Comments on the Quality of English Language

Point 1: The English is fine and does not require any improvement.

Response 1: Thank you for your comment. We are pleased to hear that the quality of the English language was found to be satisfactory. No further language editing was deemed necessary.

5. Additional clarifications

We have no further clarifications to provide. Thank you for your thorough review.